# Prevalence of Reticulocytosis in the Absence of Anemia in Dogs with Cardiogenic Pulmonary Edema Due to Myxomatous Mitral Valve Disease: A Retrospective Study

**DOI:** 10.3390/vetsci9060293

**Published:** 2022-06-14

**Authors:** Sol-Ji Choi, Won-Kyoung Yoon, Hyerin Ahn, Woo-Jin Song, Ul-Soo Choi

**Affiliations:** 1Laboratory of Veterinary Internal Medicine, College of Veterinary Medicine, Jeju National University, Jeju 63243, Korea; csj504@naver.com (S.-J.C.); gpflsl77@naver.com (H.A.); 2Guardian Angel Veterinary Hospital, Anyang 14112, Korea; cardiovetkorea@gmail.com; 3The Research Institute for Veterinary Science, College of Veterinary Medicine, Jeju National University, Jeju 63243, Korea; 4Department of Veterinary Clinical Pathology and Bio-Safety Research Institute, College of Veterinary Medicine, Chonbuk National University, Jeonju 54596, Korea

**Keywords:** cardiogenic pulmonary edema, dog, hematology, myxomatous mitral valve disease, reticulocytosis in absence of anemia

## Abstract

Myxomatous mitral valve disease (MMVD) is the most common heart disease in small breed dogs. Dogs with MMVD commonly show clinical signs of dyspnea due to cardiogenic pulmonary edema (CPE). Reticulocytosis in the absence of anemia (RAA) is a hematological finding in hypoxic conditions. We aimed to assess the prevalence of RAA in dogs with CPE due to MMVD, and evaluate whether RAA is reversible with amelioration of dyspnea. Twenty-nine client-owned dogs with CPE due to MMVD were included. Dogs who died within 6 weeks of the onset of CPE were included in the non-survival group, while the others comprised the survival group. Of the 21 dogs, RAA was observed in 17 dogs (80.9%). In the RAA group, the absolute reticulocyte count significantly decreased as CPE resolved (*p* < 0.001). The mean absolute reticulocyte count in the RAA group was 163.90 ± 50.77 on the first measurement and 78.84 ± 25.64 after resolution of CPE. In the RAA group, no significant differences in mean absolute reticulocyte count were observed between the survival and non-survival groups at either the first or second measurement. Our results indicate that RAA occurs in dogs with MMVD-related CPE and can resolve after resolution of CPE.

## 1. Introduction

Myxomatous mitral valve disease (MMVD) is the most common heart disease in small-breed dogs [1,2]. It first manifests as the regurgitation of blood flow into the left atrium by mitral valve degeneration and increased blood volume, leading to increased left atrial pressure and cardiac remodeling, and eventually resulting in increased pulmonary venous pressure and pulmonary edema (PE) [3,4]. The occurrence of PE is confirmed using thoracic radiography and echocardiography. According to the MMVD classification of the American College of Veterinary Internal Medicine (ACVIM), the onset of clinical signs is classified as stage C or higher [1].

Reticulocytes are immature erythrocytes (non-nucleated cells) that containing long RNA molecules. Reticulocytes remain for about 2 days in bone marrow and are then released into the blood circulatory system and differentiate into mature erythrocytes about 2 days later [5]. Reticulocytosis, increased reticulocyte count in the blood, increases erythrocyte production and is a quantitative indicator to distinguish regenerative anemia from non-regenerative anemia.

Reticulocytosis in the absence of anemia (RAA) refers to an increased reticulocyte count in non-anemic conditions [6,7]. Previous studies have revealed that physiologic reaction to excitement, splenic contraction, or inflammatory response could increase reticulocyte numbers in non-anemic conditions [7,8,9]. In addition, there have been many studies about hypoxic RAA [6,10,11,12]. Some studies have shown that patients with RAA have poorer prognosis [6,12]. In light of the growing interest in RAA, the study of hypoxic RAA in specific diseases would be a substantial contribution to the field of veterinary medicine [7,13]. In this study, we aimed to assess the prevalence of RAA in dogs with CPE due to MMVD, and to evaluate whether RAA is reversible with the amelioration of dyspnea.

## 2. Materials and Methods

### 2.1. Case Selection

For this retrospective case series study, we reviewed the electronic medical records of all dogs with cardiogenic pulmonary edema (CPE) due to MMVD presenting at the emergency services of our hospital, from February 2017 to August 2020.

The exclusion criteria were as follows: moderate to severe anemia, dogs that did not undergo CBC analysis within 2 days of the first hospital visit, and dogs lost to follow-up for CBC evaluation.

### 2.2. Diagnosis and Treatment of Cardiogenic Pulmonary Edema in Dogs

The diagnosis of CPE was based on a combination of clinical signs (abnormal lung sounds, dyspnea) and radiographic findings (i.e., increased pulmonary opacity resulting from unstructured interstitial or mixed interstitial alveolar pattern) on the day of the visit.

Echocardiography was performed to identify dogs with congestive heart failure secondary to MMVD [1,14,15,16]. Echocardiographic findings were characterized by degenerative changes in the mitral valve leaflets, mitral valve prolapse, and the presence of systolic mitral regurgitant flow. Each patient underwent a complete echocardiographic examination, which include left ventricular internal diameter in diastole (LVIDD), left atrial-to-aortic ratio (LA/Ao), and flow data, including peak velocity of E wave of transmitral flow, and E wave to A wave ratio of transmitral flow. LVIDD was evaluated in M-mode. LVIDDn (normalized LVIDD) was measured using the following formula: LVIDDn = LVIDD (cm)/weight (kg)^0.294^. The LA/Ao ratio was estimated from the right parasternal two-dimensional short-axis view.

CPE with MMVD was defined as the presence of heart murmur and evidence of left cardiomegaly, defined as LA/Ao ≥ 1.6 and LVIDDn > 1.7, with clinical signs based on the ACVIM continuous rate infusion classification scheme [5].

During treatment admission, all CPE dogs were initially administered oxygen via an oxygen hood. Next, furosemide was administered intravenously (2–4 mg/kg/h), and nitroglycerin patches were applied. Additionally, dobutamine was administered if needed (continuous rate infusion at 5–7.5 µg/kg/min).

### 2.3. RAA Determination

In the present study, reticulocytosis in the absence of anemia was defined as an absolute reticulocyte count greater than 100.0 K/μL and a hematocrit greater than 35%. The EDTA (ethylenediamine tetraacetic acid)-plasma samples were used for CBC analysis (in-house Procyte Dx Hematology Analyzer, IDEXX Laboratories, Cumberland County, ME, USA).

### 2.4. Follow-Up Data

For follow up, the second measurement of the CBC analysis was performed an average of 14 days (range; 10–31 days) after the first visit (except one case that died before the second visit). Patients who died within 6 weeks after CPE were included in the non-survival group, while the others comprised the survival group.

### 2.5. Statistical Analysis

Numerical data (age and absolute reticulocytes) are shown as mean ± standard deviation. Differences between the groups were compared by paired *t*-test using GraphPad Prism v.6.01 software (GraphPad Inc., La Jolla, CA, USA). Statistical significance was set at *p* < 0.05.

## 3. Results

Twenty-nine dogs were diagnosed with CPE based on their medical records. Two dogs were excluded because they had moderate to severe anemia at the first examination, and six were excluded because there were no data for the first 2 days after the hospital visit.

Twenty-one dogs of the following breeds met the criteria and were included in the study: Maltese (17), Shih Tzu (3), and Pomeranian (1). There were 10 males, including 8 castrated males, and 11 females, including 8 spayed females. Dogs ranged in age from 8 to 16 years; the median age was 12 ± 2.42 years. Two dogs had a previous history of CPE (Table 1).

Seventeen dogs had reticulocytosis (RAA group) and four dogs had normal reticulocyte counts (non-RAA group) at the time of the CPE emergency. The proportion of dogs with RAA in this study was 80.9% (Figure 1).

CPEs have resolved in 20 of 21 dogs with appropriate treatment. All remained dogs in the second visit showed normal sleeping or resting respiratory rate and no other respiratory signs. In the RAA group, the absolute reticulocyte count was decreased significantly (*p* < 0.001) at the second visit. The mean absolute reticulocyte count in the RAA group was 163.90 ± 50.77 on the first measurement and 78.84 ± 25.64 on the second measurement (Figure 2A). All the dogs had lower values than the normal reference range in the second measurement except two; one had slightly increased values, and another did not undergo the second measurement due to euthanasia on the day after the visit. In contrast, in the non-RAA group, no significant change was noted. The mean absolute reticulocyte count was 60.10 ± 15.1 on the first measurement and 68.73 ± 31.51 on the second measurement (Figure 2B). In addition, in all cases, there were no significant differences between two measurements in RBC counts (data not shown).

One dog had increased sleeping and resting respiratory rates starting 3 days before the hospital visit. Thoracic radiography revealed remarkable alveolar patterns in the right lateral and ventrodorsal views (Figure 3A). A diagnosis of MMVD with ACVIM stage C was made using echocardiography. Oxygen supplementation and furosemide intravenous injection (to provoke dehydration) were administered until body weight was reduced by 7%, and a nitroglycerin patch was applied as an emergency treatment. When discharged from the hospital, pimobendan, enalapril, furosemide, and spironolactone were prescribed. After one month, pulmonary infiltration was much improved, and breathing appeared comfortable (Figure 3B). The absolute reticulocyte count decreased from 261.8 K/μL on the first measurement to 96.7 K/μL on the second measurement (Figure 3C).

Four dogs died due to recurrent CPE within 6 weeks after the first visit, one of which was euthanized the day after the visit, and all were RAA cases. The mean absolute reticulocyte count at the first measurement was 161.09 ± 50.80 in the survival group and 188.30 ± 34.32 in the non-survival group. At the second measurement, the count was 79.91 ± 26.60 in the survival group and 85.20 ± 16.61 in the non-survival group. There were no significant differences in absolute reticulocyte count between the survival group and the non-survival group (Figure 4).

## 4. Discussion

Reticulocytosis in the absence of anemia is a clinical pathologic sign associated with a recent anemia, hemolysis, or hypoxia [17]. Recently, a study found that the prevalence of RAA in dogs is gradually increasing annually and that hypoxia is a minor cause of RAA [7]. Later, however, using a larger sample, Fuchs found that high rates of RAA in dogs were due to heart disease or respiratory disease (i.e., hypoxia) [13]. MMVD is the most common heart disease in small-breed dogs and can cause respiratory signs of hypoxia and pulmonary edema progression [13]. The aim of this study was to assess the prevalence of RAA in dogs with CPE due to MMVD and to evaluate whether RAA is reversible with the amelioration of dyspnea.

In a total sample of 21 dogs with CPE due to MMVD, the prevalence of RAA was 80.9%. This proportion was higher than the prevalence reported in a previous study (less than 10%) and a prevalence among healthy dogs (2%) [13]. There was no difference between the RAA and non-RAA groups in terms of age, reoccurrence, sex, and breed. As the respiration pattern and pulmonary infiltration improved, the RAA condition was significantly resolved in the RAA group. Based on previous studies and these results, it is tempting to speculate that RAA occurred because of hypoxia [18]. Only one dog did not show a decreased absolute reticulocyte count despite showing improvements in respiration, but we could not find the cause of this because there were no medical records with information on individual underlying disease, diet, or life patterns.

If hypoxia occurs, reticulocytes in blood increase on days 2–4 and peak on days 4–7 [17]. The peritubular interstitial cells in the renal cortex detect decreased oxygen partial pressure and inactivate oxygen-dependent prolyl hydroxylase. Thus, undestroyed hypoxia-inducible factors stimulate erythropoietin production. Erythropoietin stimulates erythroid progenitor cell division and differentiation in the bone marrow to increase erythropoiesis, which is one of the mechanisms of hypoxic RAA occurrence [5,17,19,20].

In human medicine, it is widely accepted that respiratory patients with RAA have a more negative prognosis than those without RAA [6,9]. Fuchs’ study indicated that deceased dogs with RAA had a 30% mortality rate [13]. In the present study, all deceased dogs were in the RAA group, but the correlation between absolute reticulocyte count at admission and survival was not significant.

There are some limitations in this study. We could not evaluate other possible causes of RAA such as inflammation and castration status. In addition, pulse oximetry, respiratory function (such as ventilator ratio), erythropoietin concentration, and C-reactive protein of enrolled dogs have not been measured in this study. Although these limitations are due to the retrospective nature, we demonstrated here a high prevalence of RAA in dogs with CPE due to MMVD, which was reversely downregulated.

## 5. Conclusions

To our knowledge, this is the first study to assess RAA in dogs with CPE due to MMVD. A study with a larger sample and longer study period is necessary to determine whether RAA can be used as a prognostic factor for CPE emergencies.

## Figures and Tables

**Figure 1 vetsci-09-00293-f001:**
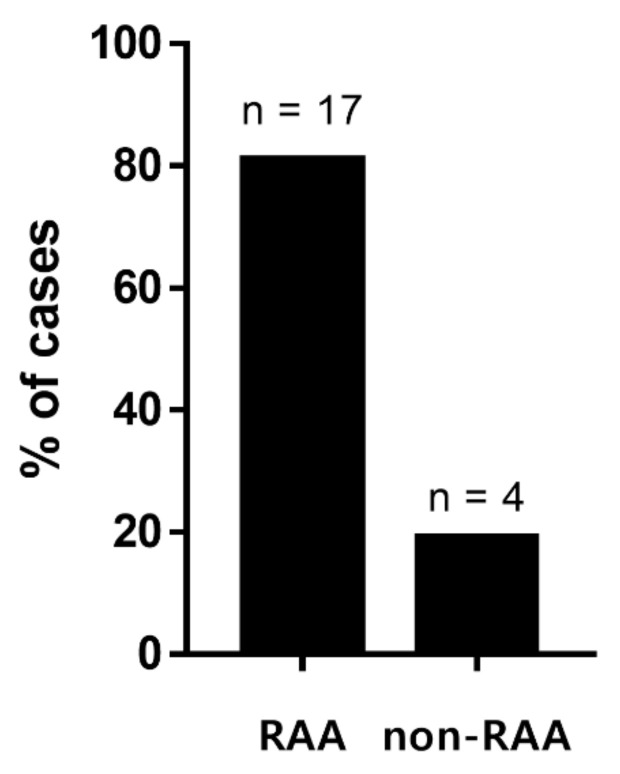
Proportion of reticulocytosis in the absence of anemia (RAA).

**Figure 2 vetsci-09-00293-f002:**
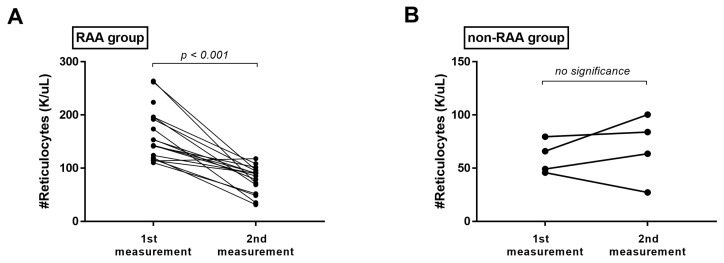
Changes in absolute reticulocyte count in (**A**) the RAA group and (**B**) the non-RAA group. RAA: reticulocytosis in absence of anemia.

**Figure 3 vetsci-09-00293-f003:**
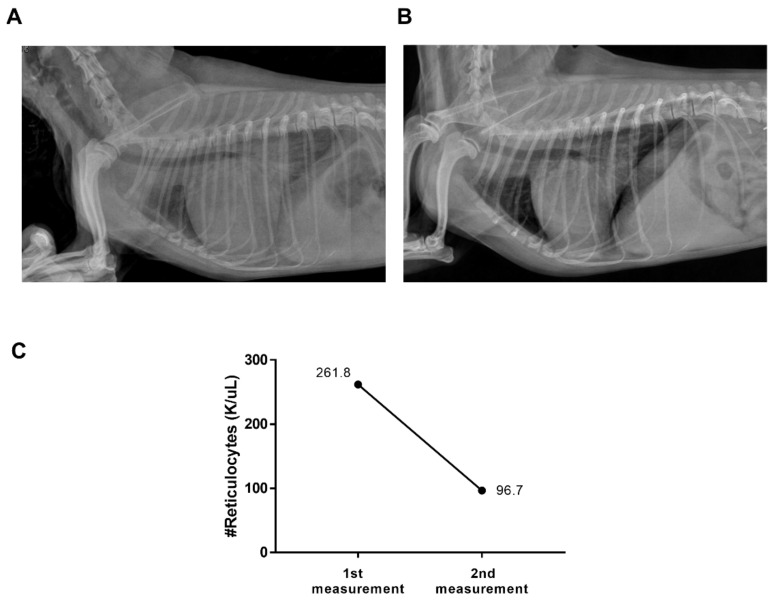
Representative case of this study. (**A**) Right lateral view of a radiography taken in an emergency visit due to cardiogenic pulmonary edema (CPE) with myxomatous mitral valve disease (MMVD). (**B**) Right lateral view of a radiography of the same case after resolution of CPE due to MMVD. (**C**) The absolute reticulocyte was decreased from 261.8 K/μL to 96.7 K/μL.

**Figure 4 vetsci-09-00293-f004:**
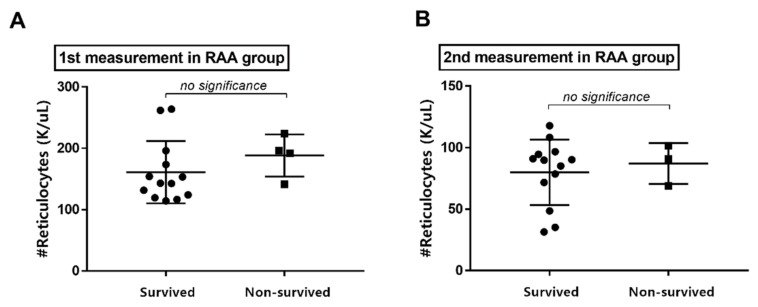
Absolute reticulocyte count in the survived and non-survived dogs in the RAA group. At both (**A**) the first and (**B**) the second measurements, there were no significant differences in absolute reticulocyte count between survived and non-survived groups. RAA: reticulocytosis in the absence of anemia.

**Table 1 vetsci-09-00293-t001:** Signalment of MMVD dogs with CPE (*n* = 29).

Characteristic	Value
Age (Mean ± SD years, range)	12 ± 2.42, 8–16
Reoccurrence (*n*, %)	2, 9.54
Sex	
Female (*n*, %)	3, 14.2
Spayed female (*n*, %)	8, 38.0
Male (*n*, %)	2, 9.5
Castrated male (*n*, %)	8, 38.0
Breed	
Maltese (*n*, %)	17, 80.9
Shih Tzu (*n*, %)	3, 17.6
Pomeranian (*n*, %)	1, 4.7

CPE: cardiogenic pulmonary edema; MMVD: myxomatous mitral valve disease.

## Data Availability

The data presented in this study are contained within the article.

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
