# Peer review of "Prevalence of Reticulocytosis in the Absence of Anemia in Dogs with Cardiogenic Pulmonary Edema Due to Myxomatous Mitral Valve Disease: A Retrospective Study"

_vetsci, 2022, doi:10.3390/vetsci9060293_

Round 1

Reviewer 1 Report

The authors reported reticulocyte numbers in 29 canine patients with cardiogenic pulmonary edema (CPE) from myxomatous mitral valve disease at two time periods and noted an increased incidence of reticulocytosis in the absence of anemia (RAA) which decreased in a subsequent measurement (with CPE treatment. This finding is interesting and the manuscript is well organized.

The findings in the paper are compelling but the study has a number of limitations. Based on the retrospective nature of the study additional necessary data may not be feasible. As a result the authors have made assertions and conclusions that may overstate the process involved. With less speculation and discussion of these limitations, and a few changes this manuscript has a high chance of publication.

Throughout the manuscript (particularly lines 18-20 and 48-50) the authors focus heavily on RAA as an indication or hypoxia. Hypoxia is just one factor that can explain RAA. Both statements should be reworded and other causes of RAA (acute responses, splenic contractions in anemic patients, inflammatory disease, and cancer) should be expanded upon in the introduction and discussion.

Pulmonary edema has also been associated with cytokine storms. RAA may be secondary to acute inflammation. Was there evidence of inflammation on the CBC of any of these patients? This should be discussed.  

Authors should include RBC counts at both time measurement times. The authors should discuss how RBC differed (if at all) in patients between the two time periods. Chronicity of MMVD and CPE may also be a factor that could differentiate the cohort with and without RAA.

Measurements were repeated in the same population, therefore a paired t-test should have been performed rather than a student t-test. Statistics and results should be updated with appropriate tests.

Stage of MVD and CPE should be included including changes in CPE stage between visits to confirm that resolution/improvement occurred. If not possible, some quantification of improvement is needed and a discussion added to study limitations indicating that CPE was not quantified or assessed each time. Without assessing CPE at both time periods the authors can only speculate on the correlation between changes in CPE and RAA.

Erythropoietin level, blood pressure measurements, pulse oximetry, and respiratory/ventilatory measurements should be included at both time periods. If that is not feasible then the authors are only speculating that hypoxia is a factor. These patients may be compensating well for any hypoxia that may or may not be occurring. Without data to support this theory the authors should revise mechanism statements to indicate they are possible explanations only and the lack of confirmatory indicators should be explained in the study limitations section of the discussion.

Perhaps as interesting as the patients with RAA are the patients without RAA. This population is too small to draw any conclusions. Increasing the study size and looking at the above factors would provide insight on the lack of RAA in this population.  If not feasible, the authors should give more attention to this population and their lack of RAA in the discussion.

Castration status is known to influence erythropoiesis. This should be discussed as a limitation if data is insufficient to test the significance of reproductive status in the manuscript.

Please reword section 2.4. It reads like the 1st measurement was exactly 1 month after the 1st visit, but the 2nd measurement was 10-30 days after the 1st visit. That would make the 2nd measurement before the 1st.  

Author Response

Reviewer #1

  1. COMMENT: The authors reported reticulocyte numbers in 29 canine patients with cardiogenic pulmonary edema (CPE) from myxomatous mitral valve disease at two time periods and noted an increased incidence of reticulocytosis in the absence of anemia (RAA) which decreased in a subsequent measurement (with CPE treatment. This finding is interesting and the manuscript is well organized.

The findings in the paper are compelling but the study has a number of limitations. Based on the retrospective nature of the study additional necessary data may not be feasible. As a result the authors have made assertions and conclusions that may overstate the process involved. With less speculation and discussion of these limitations, and a few changes this manuscript has a high chance of publication.

RESPONSE: Thank you for your kind responses and suggestions. The manuscript has certainly benefited from the reviewer’s suggestions.

  1. COMMENT: Throughout the manuscript (particularly lines 18-20 and 48-50) the authors focus heavily on RAA as an indication or hypoxia. Hypoxia is just one factor that can explain RAA. Both statements should be reworded and other causes of RAA (acute responses, splenic contractions in anemic patients, inflammatory disease, and cancer) should be expanded upon in the introduction and discussion.

RESPONSE: Thank you for your valuable comment. We have revised and added reference in the introduction section as follow: Reticulocytosis in absence of anemia (RAA) refers to an increased reticulocyte count in non-anemic conditions [Kendall, R.G.et al., Patients with Pulmonary and Cardiac Disease Show an Elevated Proportion of Immature Reticulocytes. Clin. Lab. Haematol. 2001; Pattullo, K.M. et al., Reticulocytosis in Nonanemic Dogs: Increasing Prevalence and Potential Etiologies. Vet. Clin. Pathol. 2015]. Previous studies have revealed that physiologic reaction to excitement, splenic contraction, or inflammatory response could increase reticulocyte numbers in non-anemic condition [Pattullo, K.M. et al., Reticulocytosis in Nonanemic Dogs: Increasing Prevalence and Potential Etiologies. Vet. Clin. Pathol. 2015; Stewart, I. B. et al., The human spleen during physiological stress. Sports Med. 2002; Horvath, S.J. et al., Effects of racing on reticulocyte concentrations in Greyhounds, Vet. Clin. Pathol. 2014].

We hope are approach acceptable.

  1. COMMENT: Pulmonary edema has also been associated with cytokine storms. RAA may be secondary to acute inflammation. Was there evidence of inflammation on the CBC of any of these patients? This should be discussed.

RESPONSE: Thank you for your valuable comment. As your comment, we could not evaluated other possible causes of RAA such as inflammation and castration status. In addition, Pulse oximetry, respiratory function (such as ventilator ratio), erythropoietin concentration, or C-reactive protein of enrolled dogs have not been measured in this study. Although these limitations due to retrospective nature, we demonstrated here a high prevalence of RAA in dogs with CPE due to MMVD, which was reversely down-regulated. It has been described in discussion section. We hope our approach acceptable.

  1. COMMENT: Authors should include RBC counts at both time measurement times. The authors should discuss how RBC differed (if at all) in patients between the two time periods. Chronicity of MMVD and CPE may also be a factor that could differentiate the cohort with and without RAA.

RESPONSE: Thank you for your detailed comment, and we agree with you. In all cases, there were no significant differences between two measurements in RBC counts (data not shown). It has been described in the revision.

  1. COMMENT: Measurements were repeated in the same population, therefore a paired t-test should have been performed rather than a student t-test. Statistics and results should be updated with appropriate tests.

COMMENT: Thank you for your detailed comment, and we agree with you. Differences between the groups were compared by paired t-test using GraphPad Prism v.6.01 software (GraphPad Inc., La Jolla, CA, USA). It has been described in the revision.

  1. COMMNET: Stage of MVD and CPE should be included including changes in CPE stage between visits to confirm that resolution/improvement occurred. If not possible, some quantification of improvement is needed and a discussion added to study limitations indicating that CPE was not quantified or assessed each time. Without assessing CPE at both time periods the authors can only speculate on the correlation between changes in CPE and RAA.

RESPONSE: Thank you for your valuable comment. In this study, CPE have resolved in 20 of 21 dogs with appropriate treatment. All remained dogs in the second visit have showed normal sleeping or resting respiratory rate and no any other respiratory sign. And these results from remained dogs are consistent with MMVD ACVIM stage Cc (without CPE) at the time of second measurement. However, 4 dogs died due to recurrent CPE within 6 weeks after the first visit, one of which was euthanized the day after the visit, and all were RAA cases. It has been described in the revision (results section). We hope our approach acceptable.

  1. COMMENT: Erythropoietin level, blood pressure measurements, pulse oximetry, and respiratory/ventilatory measurements should be included at both time periods. If that is not feasible then the authors are only speculating that hypoxia is a factor. These patients may be compensating well for any hypoxia that may or may not be occurring. Without data to support this theory the authors should revise mechanism statements to indicate they are possible explanations only and the lack of confirmatory indicators should be explained in the study limitations section of the discussion.

COMMENT: : Thank you for your valuable comment. As your comment, we could not evaluated other possible causes of RAA such as inflammation and castration status. In addition, Pulse oximetry, respiratory function (such as ventilator ratio), erythropoietin concentration, or C-reactive protein of enrolled dogs have not been measured in this study. Although these limitations due to retrospective nature, we demonstrated here a high prevalence of RAA in dogs with CPE due to MMVD, which was reversely down-regulated. It has been described in discussion section. We hope our approach acceptable.

  1. COMMENT: Perhaps as interesting as the patients with RAA are the patients without RAA. This population is too small to draw any conclusions. Increasing the study size and looking at the above factors would provide insight on the lack of RAA in this population. If not feasible, the authors should give more attention to this population and their lack of RAA in the discussion.

RESPONSE: Thank you for your valuable comment, and we agree with you. We have revised the manuscript (conclusion section) as follow: To our knowledge, this is the first study to assess RAA in dogs with CPE due to MMVD. A study with a larger sample and longer study period is necessary to determine whether RAA could be used as a prognostic factor for CPE emergencies.

We hope our approach acceptable.

  1. COMMENT: Castration status is known to influence erythropoiesis. This should be discussed as a limitation if data is insufficient to test the significance of reproductive status in the manuscript.

RESPONSE: Thank you for your valuable comment. As your comment, we could not evaluated other possible causes of RAA such as inflammation and castration status. In addition, Pulse oximetry, respiratory function (such as ventilator ratio), erythropoietin concentration, or C-reactive protein of enrolled dogs have not been measured in this study. Although these limitations due to retrospective nature, we demonstrated here a high prevalence of RAA in dogs with CPE due to MMVD, which was reversely down-regulated. It has been described in discussion section. We hope our approach acceptable.

  1. COMMENT: Please reword section 2.4. It reads like the 1st measurement was exactly 1 month after the 1st visit, but the 2nd measurement was 10-30 days after the 1st visit. That would make the 2nd measurement before the 1st.

RESPONSE: Thank you for your detailed comment, and it was our mistake. We have revised the manuscript as follow: For follow up, the second measurement of the CBC analysis was performed on an average of 14 days (range; 10–31 days) from the first visit (except one case that died before the second visit).

We hope our approach acceptable.

Reviewer 2 Report

Authors assess the prevalence of reticulocytosis in absence of anemia in dogs with cardiogenic pulmonary edema due to myxomatous mitral valve disease, and evaluate whether RAA is reversible with amelioration of dyspnea.

The title indicates the aim of the manuscript and the abstract is well written. The introduction is also well written. The objectives of the study are of interest and are in line with the scope of the journal.

The manuscript is well organized. The abstract clearly indicates the work objective, methodology and result of the study. The methodology is well articulated and the description is well made.

The conclusions are consistent with the evidence and arguments presented.

The reference is appropriate.

In my opinion, the manuscript could be accepted for publication.

Author Response

Reviewer #2

  1. COMMENT: Authors assess the prevalence of reticulocytosis in absence of anemia in dogs with cardiogenic pulmonary edema due to myxomatous mitral valve disease, and evaluate whether RAA is reversible with amelioration of dyspnea.

The title indicates the aim of the manuscript and the abstract is well written. The introduction is also well written. The objectives of the study are of interest and are in line with the scope of the journal.

The manuscript is well organized. The abstract clearly indicates the work objective, methodology and result of the study. The methodology is well articulated and the description is well made.

The conclusions are consistent with the evidence and arguments presented.

The reference is appropriate.

In my opinion, the manuscript could be accepted for publication.

RESPONSE: Thank you for your time and effort for reviewing our manuscript. Also, we thanks for your kind response.